# Effects of PROtein enriched MEDiterranean Diet and EXercise on nutritional status and cognition in adults at risk of undernutrition and cognitive decline: the PROMED-EX Randomised Controlled Trial.

Nicola Ann Ward ,[1] Rachel Reid-McCann,[1] Lorraine Brennan,[2] Christopher R Cardwell,[1] CPGM de Groot,[3] Stefania Maggi,[4] Noel McCaffrey,[5] Bernadette McGuinness,[1] Michelle C McKinley,[1] Marianna Noale,[4] Roisin F O'Neill,[1] Federica Prinelli,[6] Giuseppe Sergi,[7] Caterina Trevisan,[7] Dorothee Volkert,[8] Jayne V Woodside,[1] Claire T McEvoy[1,9]

For numbered affiliations see end of article.

**Correspondence to**
Dr Claire T McEvoy;
c.mcevoy@qub.ac.uk

## ABSTRACT

**Introduction** Undernutrition leading to unplanned weight loss is common in older age and has been linked to increased dementia risk in later life. Weight loss can precede dementia by a decade or more, providing a unique opportunity for early intervention to correct undernutrition and potentially prevent or delay cognitive impairment. The combined effects of diet and exercise on undernutrition have not yet been evaluated. The objective of this trial is to determine the effect of a protein-enriched Mediterranean diet, with and without exercise, on nutritional status and cognitive performance in older adults at risk of undernutrition and cognitive decline.

**Methods** One hundred and five participants aged 60 years and over at risk of undernutrition and with subjective cognitive decline will be recruited to participate in a 6-month, single-blind, parallel-group randomised controlled trial. Participants will be block randomised into one of three groups: group 1—PROMED-EX (diet+exercise), group 2—PROMED (diet only) and group 3—standard care (control). The primary outcome is nutritional status measured using the Mini Nutritional Assessment. Secondary outcomes include cognitive function, nutritional intake, body composition, physical function and quality of life. Mechanistic pathways for potential diet and exercise-induced change in nutritional status and cognition will be explored by measuring inflammatory, metabolic, nutritional and metabolomic biomarkers.

**Ethics and dissemination** The study is approved by the UK Office for Research Ethics Committee (ref: 21/NW/0215). Written informed consent will be obtained from participants prior to recruitment. Research results will be disseminated to the public via meetings and media and the scientific community through conference presentations and publication in academic journals.

**Trial registration number** ClinicalTrials.gov Registry (NCT05166564).

## STRENGTHS AND LIMITATIONS OF THIS STUDY

⇒ This is the first randomised controlled trial designed to evaluate the effect of an optimised protein-enriched Mediterranean diet, with and without exercise, on nutritional status and cognitive performance in a high-risk population.

⇒ The integration of state-of-the-art metabolomics, metabolic, inflammatory and nutritional biomarkers will help elucidate biological mechanisms underlying how diet can influence nutrition status and cognition and how exercise can interact with nutrition to modify these processes.

⇒ A patient and public involvement group will contribute at all stages of the PROMED-EX trial to optimise the research design and dissemination strategy of the research findings.

⇒ The nature of the 'real-world' pragmatic study design means that it is not possible to blind participants to treatment allocation, possibly leading to performance bias.

⇒ Data will be analysed by a blinded member of the study team to minimise detection bias.

## INTRODUCTION

Undernutrition, defined as insufficient food and/or nutrient intake that results in altered body composition and unplanned weight loss,[1] is common in older age and affects >1.3 million older adults in the UK.[2] Individuals with cognitive impairment are particularly vulnerable to undernutrition and its many adverse consequences. Undernourished people with dementia experience more rapid decline in cognitive and physical functions, increased neuropsychiatric symptoms,

and greater risk of hospitalisation and mortality, than those with adequate nutritional status.[3 4] Being undernourished has also emerged as a potentially modifiable risk factor for dementia.[5] In prospective studies, weight loss has been associated with increased dementia risk,[6] and precedes the onset of dementia by a decade or more,[7 8] making it an intervention target that should be prioritised.

Causes of undernutrition are increased nutrient requirements, inadequate dietary intake and reduced nutrient bioavailability,[9] but the exact mechanisms of undernutrition in dementia are not clear. It is possible that weight loss may be both a consequence and cause of neurodegeneration and dementia. Loss of muscle and weight in older adults has been linked to accelerated decline in brain structures,[10–12] even after exclusion of cognitive impairment,[12] suggesting a pathophysiological role of undernutrition in dementia. In contrast, weight loss could be an early marker of dementia development. Pathological changes in the brain are present 10–15 years prior to the clinical symptoms of cognitive impairment and induce inflammatory[13] and physiological changes beyond normal ageing that can have a negative impact on nutrition and body weight. Dementia pathology affects brain regions such as the hypothalamus which is important for appetite regulation and eating behaviour.[14] This may explain alterations of leptin and ghrelin metabolic signals that are observed in early disease.[15] Early brain changes can also affect chemosensory perception including taste and olfactory function,[16] as well as satiety regulation, protein and lipid synthesis,[17] and food preferences. Disease progression and worsening cognitive impairment can cause forgetfulness to eat, dysphagia,[18] sleep disturbance[19] and social withdrawal,[20] which can all have adverse effects on food intake.[21]

To date, most studies have focused on the impact of diet on physical function,[22] with limited number of interventions targeting undernutrition for cognitive benefit. Short-term treatment of undernutrition has been shown to improve both nutritional status and cognitive performance.[23–25] However, small sample sizes as well as differences in residential status of study populations and nutritional interventions (dietetic counselling,[23] oral supplementation,[24 25] meal provision[25]), and cognitive outcomes make it difficult to know whether addressing undernutrition could protect against cognitive decline. Hence, trials are warranted to test the efficacy of intervention for undernutrition in populations at increased risk of cognitive decline but not yet cognitively impaired and may benefit from early intervention.

The Mediterranean diet (MedDiet) is a nutrient-dense dietary pattern with proven anti-inflammatory benefits[26] and may offer protection against cognitive decline during ageing. Observational data have reported links between the MedDiet and favourable changes in brain structures and functions,[27] slower rate of cognitive decline[28 29] and decreased dementia risk,[30 31] which appear to be attributable to the combined effect of different foods rather than the action of a single food or nutrient.[32] Furthermore, the MedDiet is associated with nutritional adequacy and lower risk of undernutrition and micronutrient deficiency.[33] Adequate protein (1.0–1.2/kg body weight/day)[34] and energy intake[35] are critical for preventing muscle loss[36] and undernutrition in older adults.[37] However, emerging data suggest that a high-quality dietary pattern is also important,[38] lending support to a 'whole diet' approach for preventing muscle loss and functional decline during ageing.

Physical exercise has positive effects on cognition,[39] especially in domains of executive function and processing speed,[40] by increasing neurogenesis and upregulating neurotrophic factors.[41] However, it is not known how diet and exercise interact for neuroprotection. Multidomain interventions that include both a healthy diet and exercise appear to elicit stronger positive results for cognitive performance than each intervention in isolation.[42] Furthermore, exercise can stimulate protein synthesis[43] and enhance muscle mass and functional performance in older adults,[44] making it an attractive target for preventing undernutrition. An optimal protein MedDiet may induce additive effects on nutritional status and cognition when combined with exercise, but this has not yet been evaluated.

The PROtein enriched MEDiterranean Diet and EXercise (PROMED-EX) randomised controlled trial (RCT) aims to determine the effects of a protein-enriched MedDiet, with and without exercise, in comparison with standard care, on nutritional status in adults at risk of undernutrition and cognitive decline. PROMED-EX is part of the European 'PROMED-COG' Project, funded under the Horizon Joint Programming Initiative 'A Healthy Diet for a Healthy Life' to establish evidence for the balance between diet and physical activity for preventing undernutrition and promoting healthy neurocognitive ageing.[45]

## METHODS AND ANALYSIS

PROMED-EX is a 6-month, single-blind, parallel-group RCT aiming to recruit 105 eligible adults residing in Northern Ireland (NI), aged 60+ years and at risk of both undernutrition and cognitive decline. The protocol has been written in accordance with the Standard Protocol Items: Recommendations for Interventional Trials guidelines,[46] and a summary is reported using the checklist (online supplemental appendix 1). An independent Trial Steering Committee (TSC) has been established (online supplemental appendix 2) to oversee the conduct of the study.

### Patient and public involvement

A patient and public involvement (PPI) group (n=8) comprising older adults residing in NI with lived experience of subjective cognitive decline (SCD) and/or poor nutritional status were recruited for PROMED-EX. PPI members have contributed to the intervention

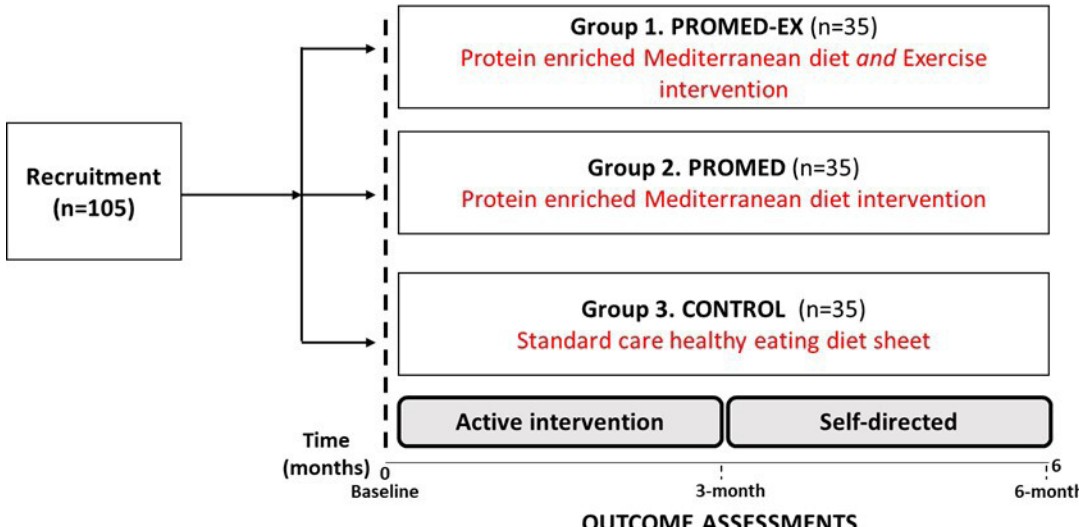

**Figure 1** Overview of PROMED-EX Study design. PROMED-EX, PROtein enriched MEDiterranean Diet and EXercise.

design and co-development of materials, including lay summary, participant information sheet and consent form, advertisement materials and written educational resources for the interventions. Besides the trial design, two PPI members sit on the TSC and will provide input to future stages including implementation, analysis and dissemination.

## Study design

Study participants will be randomised into one of three groups: group 1—PROMED-EX: protein-enriched MedDiet and exercise intervention (n=35); group 2—PROMED: protein-enriched MedDiet intervention (n=35); or group 3—control: standard care healthy eating diet sheet (n=35) as shown in figure 1. Participants will be informed of their allocated group after baseline data have been collected. Those randomised to the PROMED-EX and PROMED interventions will receive active support to adopt the behavioural targets in the first 3 months, after which time they will continue with self-directed maintenance of the newly adopted behaviours for a further 3 months.

## Sample size

As no prior studies have evaluated the effects of whole diet and exercise on nutritional status and cognition, the sample size calculation is based on an SD of the change in Mini Nutritional Assessment (MNA) score of 4 points over a similar period from the control group of a previous nutrition intervention.[47] A total of 28 completing in each intervention and control group would provide over 80% power to detect a difference in the three means, based on a statistically significant analysis of variance (ANOVA) result at the 5% level, hypothesising a true increase in MNA of 3 or more in both intervention groups, but not the control group. Loss to follow-up is estimated from our prior experience of conducting lifestyle trials in older adults[48–50] where attrition ranges from 6%[50] to 30%.[49] Considering, FINGER[42] evaluated an intensive multidomain lifestyle intervention among older adults at increased dementia risk and reported 12% withdrawal over 2 years. We anticipate dropout at 20% and therefore have added 20% to the sample size to account for non-compliance, dropout or discontinuation.

## Recruitment and eligibility of study participants

Multiple recruitment strategies will be employed including paid Facebook advertisements, other media (eg, newspaper, radio and television), and identification of potentially eligible participants from community networks, primary care sectors and health and social care trusts in NI. Those interested in participating will be provided with an information sheet detailing the study protocol, followed by a phone call to discuss study procedures.

Potentially eligible participants will be screened by telephone/video to last around 45 min. Individuals will be eligible to enrol if they are community dwelling, aged ≥60 years, at risk of undernutrition defined by an MNA-Short Form score of 8–11[51] and have a positive screen for SCD.[52] The screening assessment includes the Telephone Cognitive Screen or Mini Mental State Examination (MMSE) to exclude participants with cognitive impairment and a health and medical history questionnaire to determine if the participant has any medical conditions that compromise their ability to take part. This includes if they receive nutritional support, have a diagnosis of mild cognitive impairment or dementia, or dietary restrictions as shown in table 1.

## Randomisation, blinding and allocation

A study statistician independent of recruitment, intervention delivery and data collection will perform block randomisation using a computer-generated random number sequence in blocks of size 6 (using command RALLOC in STATA V.16, StataCorp, College Station, USA). Allocations will be prepared in sealed opaque envelopes by an independent study administrator and will not be revealed to participants until completion of baseline data

**Table 1** PROMED-EX trial eligibility criteria

| Inclusion criteria | Exclusion criteria |
|---|---|
| ► Age ≥60 years<br>► Community dwelling<br>► Mini Nutritional Assessment-Short Form score 8–11, indicating increased risk of undernutrition[51]<br>► Positive screen for subjective cognitive decline[52] defined as a 'yes' response to the question: 'During the past 12 months, have you experienced confusion or slight memory loss that is happening more often or is getting worse?'<br>► Ability to provide informed consent | ► Cognitive impairment: Telephone Cognitive Screen ≤21 or Mini Mental State Examination ≤24 or clinical diagnosis of mild cognitive impairment or dementia<br>► In receipt of prescribed oral or artificial nutritional support<br>► Dietary restrictions/allergies to PROMED foods (eg, nuts) or taking high-dose nutritional supplements (eg, vitamin E) that limit the ability to take part in the study<br>► Living in long-term care facilities<br>► Self-reported:<br> – Dysphagia<br> – Chronic kidney disease<br> – Diabetes complications or HbA1c >8%<br> – Significant medical comorbidities that may alter cognitive test performance, for example, stroke, brain injury<br>► Inability to complete cognitive testing due to language or visual impairment |

PROMED-EX, PROtein enriched MEDiterranean Diet and EXercise.

collection. It will not be possible to blind the participant nor study researchers involved in intervention delivery to the allocated intervention group; however, data collection and analysis will be carried out by investigators blinded to treatment allocation.

### Interventions
#### Group 1: PROMED-EX
##### Protein-enriched MedDiet intervention
The PROMED-EX trial will be overseen by a registered dietitian (CTM). Participants will meet a trained researcher, with a background in nutrition and exercise within 1 week following randomisation to receive personalised dietary counselling based on their baseline anthropometry, MNA score and PROMED diet score (online supplemental appendix 3). This session will last up to 60 min where education will be provided on adequate nutritional status, avoiding weight loss and encouraging regular meal pattern. Participants will be encouraged

---

**Box 1    PROMED diet behaviours**

**Daily**
⇒ 1 tablespoon of olive oil.
⇒ ≥2 portions of fruit.
⇒ ≥3 portions of vegetables.
⇒ ≥2 portions of wholegrain foods (including wholemeal bread, wholegrain rice, whole wheat pasta).
⇒ ≥2 portions of dairy foods (including, milk, cheese, yoghurts) or plant protein alternatives.
⇒ Moderate alcohol (1-2 glasses of wine/beer consumed with meals) for alcohol consumers.

**Weekly**
⇒ Preferential consumption of chicken and oily fish over red meat, ≥2 portions of each and red meat <2 portions.
⇒ 3 handfuls of natural nuts (90 g).

---

to set dietary goals to adopt PROMED diet behaviours (shown in Box 1) to achieve overall intervention targets: (1) increased protein intake up to 1.5 g/kg current body weight,[37] (2) increased MNA score by ≥2 points and (3) increased PROMED diet score by ≥2 points. Daily energy requirements will be calculated at 30 kcal/kg of body weight with a supplement of 400–600 kcal only if weight gain is indicated.[53]

Written materials (PROMED diet guide and recipe book) will be provided to support adoption of diet behaviours. Participants will be given a recipe book with suggested quick, low-cost menu plans and practical dietary tips to overcome small appetite. Key foods will be delivered to their home free of charge for the first 3 months to encourage adoption of the PROMED diet behaviours. Foods will be self-selected by participants from a predefined list to facilitate personal preference and will include olive oil, fruits and vegetables, wholegrain foods (cereals, breads, rice and pasta), nuts, nut butters, canned oily fish and low-fat dairy foods.

##### Exercise intervention
The exercise intervention has been adapted from a commercially available 'ExWell@home' group exercise rehabilitation programme that is acceptable to older adults and suggested to improve health outcomes in those with chronic disease.[54] The home-based exercise intervention comprises strength, aerobic, core stability and balance exercises available across four levels of ability: (1) chair based, (2) basic, (3) standard and (4) advanced. After the dietary education session, participants will receive a personal exercise prescription involving a written plan, demonstration of exercises suitable for their fitness level and instruction on safety aspects to minimise injury risk. A researcher trained to deliver the adapted ExWell Programme will determine baseline fitness level

using measured blood pressure, pulse rate, short physical performance test results, handgrip strength test obtained from the baseline study visit and their assessed performance in a 'basic level' class.

Participants will be asked to report their perceived intensity of exertion using the Borg Rate of Perceived Exertion Scale[6–20 55] aiming for 'somewhat hard' within a range of 12–14 points. Participants will be instructed to increase or decrease the frequency and intensity of exercise over the first 4 weeks to meet the intervention target of two 30–60 min moderate-intensity exercise sessions per week. Participants will receive additional supportive materials to achieve their exercise goals including pre-recorded exercise classes, a written home-based exercise resource, a pedometer and exercise diary.

### Group 2: PROMED

Participants will receive only the protein-enriched MedDiet intervention and will be advised to keep other lifestyle factors unchanged throughout the study period, including maintenance of their usual physical activity level.

### Group 3: standard care

Participants will receive a general healthy eating diet sheet provided at baseline. To encourage retention in the study, personalised dietary advice and written materials will be offered to control group participants upon completion of the final study assessment.

### Maximising and monitoring compliance with the diet and exercise interventions

For study participants randomised to the PROMED-EX and PROMED interventions (groups 1 and 2, respectively), several approaches will be used to facilitate behaviour change and encourage compliance with the diet and exercise goals. The initial education session will be scheduled at a convenient time for the participant and in a convenient location (either in a dedicated research facility, the participant's own home or via remote video call). Partners or caregivers who are responsible for food preparation and/or shopping will also be invited to the education session if applicable. Furthermore, techniques demonstrated to support diet and exercise behaviour change have been incorporated into the intervention written materials and include 'instruction on how to perform behaviour', 'action planning', 'self-monitoring' and 'problem-solving'.[56]

Participants will be encouraged to self-monitor their compliance in achieving the diet and exercise goals where applicable. The researcher will contact participants weekly via telephone during the first 3 months to review individual progress, monitor compliance and body weight, deal with any difficulties participants may be experiencing in meeting their goals and provide support and personalised feedback as required. Participants will also be able to contact the researcher at any time during the 6-month intervention period.

### Data collection and study outcomes

Participants will meet with a researcher at baseline and after 3 and 6 months to complete a study assessment lasting no longer than 2.5 hours. The assessment will take place in a dedicated research facility at Queen's University Belfast (QUB) or in the participant's home if preferred. A summary of the outcomes and data collection measures at study time points is found in table 2.

### Primary outcome

The primary study outcome is between-group difference in change in nutritional status at 6 months using the validated MNA score (MNA 0–30).[57] The MNA is the most widely validated tool for detecting undernutrition risk in older adults[58] and is shown to respond to dietary intervention.[47]

### Secondary outcomes

#### Nutrition and body composition

Change in MNA score will be analysed at midpoint (3 months from baseline). Dietary behaviour will be reported using the PROMED diet score (range 0–14) adapted from a validated Spanish Mediterranean diet score[59] to incorporate the higher protein food targets and tailored for a Northern European population. Higher scores indicate increased concordance with the PROMED diet. Overall dietary intake will be assessed using a 4-day prospective food diary. Participants will be instructed to record all items of food and beverages consumed over 4 consecutive days (2 weekend days) at home and provide details of the portion size consumed, cooking methods and named brands. Data on self-reported appetite will also be collected using the four-item Simplified Nutrition Appetite Questionnaire (SNAQ) validated for older community-dwelling adults.[60] Lower SNAQ scores indicate poorer appetite. Body composition will be assessed using non-invasive bioelectrical impedance analysis (BodyStat 1500 MDD) to estimate body water, fat and lean muscle mass.[61] Anthropometric measures of weight, height and waist circumference will also be taken using digital scales, a stadiometer and flexible measuring tape, respectively.

#### Physical activity

Time spent on different intensity levels of physical activity will be self-reported using a validated Recent Physical Activity Questionnaire across four domains: home, commuting, occupational activity (if appropriate) and leisure activity over the previous 4 weeks.[62]

#### Cognitive function

A brief neuropsychological test battery (NTB) will be administered by a trained researcher to capture cognitive domains that are shown to be sensitive to progression in early stages of memory decline: global cognition, executive functioning, episodic and working memory, information processing speed and sustained attention.[63] The NTB will include the Catch-Cog assessment[63] comprising (1) three Alzheimer's Disease Assessment Scale–Cognitive

**Table 2** Overview of study data collection variables and measures

| Study outcomes | Data collection variables/measures | Baseline | 3 months | 6 months |
|---|---|---|---|---|
| Primary outcome | | | | |
| Nutritional status (6 months) | Mini Nutrition Assessment (MNA) Long-Form Questionnaire[57] | ✓ | | ✓ |
| Secondary outcomes | | | | |
| Nutrition and body composition | | | | |
| Nutritional status (midpoint change) | MNA Long-Form | ✓ | ✓ | |
| Nutritional intake | PROMED diet score questionnaire and 4-day food diary | ✓ | ✓ | ✓ |
| Appetite | Simplified Nutrition Appetite Questionnaire[60] | ✓ | ✓ | ✓ |
| Body composition | Bioelectric impedance analysis of body water, fat and lean muscle mass (BodyStat 1500 MDD) | ✓ | ✓ | ✓ |
| Anthropometry | Height (stadiometer) and weight (digital scales); waist circumference | ✓ | ✓ | ✓ |
| Physical activity | | | | |
| Physical activity | Recent Physical Activity Questionnaire[62] | ✓ | ✓ | ✓ |
| Cognitive function | | | | |
| Cognitive function | Catch-Cog,[63] Trail Making Test A and B,[64] and Delayed Recall Test | ✓ | ✓ | ✓ |
| Physical strength and function | | | | |
| Functional capacity | Short Physical Performance Battery[65] | ✓ | ✓ | ✓ |
| Muscle strength | Hand Grip Strength hydraulic dynamometer (Baseline) | ✓ | ✓ | ✓ |
| Psychosocial | | | | |
| Health-related quality of life | SF-36 Health Survey[66] Questionnaire | ✓ | ✓ | ✓ |
| Depressive symptoms | Geriatric Depression Scale[67] Questionnaire | ✓ | ✓ | ✓ |
| Blood pressure | | | | |
| Blood pressure | Clinic-measured blood pressure (Omron Healthcare) | ✓ | ✓ | ✓ |
| Biochemical measures | | | | |
| Blood biomarkers | Nutritional: vitamins A and E, and carotenoids<br>Cardiometabolic: lipid profile, HbA1c, glucose and insulin<br>Inflammatory: hs-CRP, IL-6<br>Metabolomic analysis | ✓ | ✓ | ✓ |
| Evaluation | | | | |
| Study evaluation | Questionnaire | | | ✓ |

hs-CRP, high-sensitivity C reactive protein; IL, interleukin.

subscales (Word Recognition (score range 0–12), Word Recall (score range 0–10) and Orientation (score range 0–8)) to assess episodic memory; (2) the Controlled Oral Word Association Test (number of acceptable words in 1 min) and Category Fluency Test (number of animals named in 1 min); (3) Digit Symbol Substitution Test (total number of correct symbols in 90 s); and (4) Digit Span Backward (score range 0–14) to assess executive function and working memory. The Trail Making Test (A and B) will be used to assess information processing speed and attention.[64] A delayed recall test of recalled learnt words after a 5-minute delay (score range 0–10) will be administered, with higher scores indicating better memory performance. Summary composite scores will be created for global cognition and domains of memory and executive function.

### Physical strength and function

Lower extremity functional status will be measured using the Short Physical Performance Battery (SPPB).[65] The SPPB includes three timed tasks to assess walking speed, chair stand and standing balance, with scores ranging from 0 (worst performance) to 12 (best performance). Muscle (handgrip) strength will be measured with a calibrated dynamometer (Baseline) (kg) using a standard procedure.[48]

## Psychosocial

Health-related quality of life (HRQoL) will be determined by using the Short-Form 36-item Questionnaire (SF-36),[66] which measures eight domains of health status: physical functioning, role limitations due to physical issues, role limitation due to emotional problems, energy/fatigue, emotional well-being, social functioning, pain and general health perception. SF-36 scores range from 0 to 100 with higher scores indicating better HRQoL.[66] Depressive symptoms will be determined using the 15-item Geriatric Depression Scale (GDS)–Short-Form. GDS scores range from 0 to 15, with a score ≥5 suggesting depression.[67]

## Blood pressure

Systolic and diastolic blood pressure (mm Hg) will be measured three times in the dominant arm after a 5-minute rest in the seated position using a calibrated automated device approved by the British Hypertension Society (Omron Healthcare). The first measure will be discarded and the average of the final two measurements will be recorded.

## Biochemical measures

A 50 mL fasting peripheral venous blood sample will be taken at the start of each study assessment for metabolomic profiling and biomarker measurement. Anonymised samples will be processed in the nutrition laboratory at QUB according to standardised protocols and stored in aliquots at −80°C until analysis. Biochemical markers relating to nutrition (vitamins A, E and carotenoids), metabolic (lipid profile, HbA1c, glucose and insulin) and inflammation (high-sensitivity C reactive protein, interleukin-6) will be measured. Plasma samples will be transferred to University College Dublin for metabolomic analysis.

## Evaluation

A questionnaire will be administered at the end of the study to determine acceptability and tolerance of the intervention and the overall experience of taking part in the study. Participants will be asked to return diet and exercise logs to evaluate engagement with the intervention.

## Data management and statistical analysis plan

Data will be de-identified and stored securely according to General Data Protection Regulation and UK Data Protection Act 2018 in QUB. There are no plans for interim study analysis. The groups will be similar due to randomisation; therefore, there are no a priori plans to include covariates in the analyses. Participant characteristics by allocated group will be compared using ANOVA for continuous variables and $\chi^2$ test for dichotomous variables. Skewed variables will be log-transformed prior to analysis. An intention-to-treat analysis will be conducted using multiple linear regression model to compare MNA score at 6 months between the three groups adjusting for baseline. Dunnett's procedure will then be used to compare each intervention group to the controls, accounting for multiple testing. Analysis of secondary outcomes will use a similar method and these exploratory analyses will be interpreted cautiously. Sensitivity analyses will be conducted to evaluate the diet and exercise behavioural changes maintained over 6 months and to determine the effect of intervention compliance on study endpoints. Multiple imputations for missing data will be used if required. Analyses will be performed using STATA Statistical Software (StataCorp).

## Ethics and dissemination

Ethical approval has been received from the Office for Research Ethics Committee (OREC) (ref: 21/NW/0215) and the trial has been registered on ClinicalTrials.gov (NCT05166564). The study will be reported according to Consolidated Standards of Reporting Trials guidelines for parallel-group randomised trials.[68] Written informed consent will be obtained from all study participants prior to study enrolment and any adverse events will be reported to the study sponsor (QUB) and OREC.

Study findings will be of interest to a range of stakeholders in the areas of healthy ageing, preventable malnutrition, cognitive decline and dementia risk reduction, including academics, the public, healthcare professionals and clinicians, charitable organisations and industry. Findings will be published in academic journals and published in open-access repositories, for example, the JPI-HDHL Meta Database (https://www.healthydietforhealthylife.eu/) knowledge transfer platform.

## COVID-19

The research team will comply with public health and QUB guidance regarding COVID-19. Several strategies will be implemented to mitigate risk: (1) COVID-19 risk screening for participants prior to study visits, (2) optional home study visits, (3) optional remote delivery of intervention support, (4) personal protective equipment for researchers conducting study assessments, (5) participant face coverings for study visits, (6) social distancing measures followed where possible and (7) researcher lateral flow testing prestudy visit.

## DISCUSSION

To our knowledge, PROMED-EX is the first study aimed to evaluate the combined effects of a protein-enriched MedDiet and exercise on nutritional status and cognition in older adults at risk of undernutrition with SCD. Lifestyle interventions to address modifiable risk factors for dementia are likely to have a more significant public health impact when implemented at an early stage of disease and in high-risk populations. PROMED-EX will generate new knowledge about how a targeted lifestyle intervention delivered in a 'real-world' setting can impact nutritional and cognitive status among older individuals at risk of undernutrition and cognitive decline. A further novel aspect is the planned evaluation

that will provide insight into lifestyle behaviour changes that are feasible and acceptable for the intended population.

A key strength is the inclusion of a validated neurocognitive test battery in the current study allowing a comprehensive assessment of performance in both memory and non-memory domains of cognition as used in other multidomain interventions in adults at increased dementia risk.[42] Prior studies in undernourished patients have relied on the limited MMSE global cognition to measure cognitive performance.[25] While MMSE is a validated screener for cognitive impairment, it provides limited examination of visuospatial cognitive ability, long-term memory and language functions. It also has a 'ceiling' effect whereby individuals can still score quite highly even though they may have significant cognitive impairment.[69]

A further strength is the integration of state-of-the-art metabolomics, metabolic, inflammatory and nutritional biomarkers in the study will help to elucidate biological mechanisms underlying how diet can influence nutrition status and cognition, and how exercise may act to modify these processes. Furthermore, the interventions will be delivered using a personalised approach with incorporation of evidence-based techniques to support and encourage behaviour change. Study participants will also benefit from written educational resources that have been co-designed with PPI members to support the desired diet and exercise behaviour changes. The ongoing collaboration with PPI members in the trial will undoubtedly optimise the dissemination of study findings.

There are some limitations to the study design that should be considered. The optimum period for lifestyle-induced change in nutritional status is not clear, and the duration of the current study intervention is shorter than in other multidomain trials[42]; however, improvement in MNA score in response to nutrition intervention has been observed in 12 weeks.[65] Furthermore, 6 months should be sufficient duration to detect increases in nutrient biomarkers in response to the interventions, especially in a high-risk population.

The pragmatic study design means that we may have limited power to detect between intervention group effects on MNA. Furthermore, due to the novelty of the intervention and population, it was not possible to estimate sample size calculation for cognition; however, the findings generated from PROMED-EX on cognitive performance will be used to inform future studies.

The level of research support for lifestyle behaviour change differs between the intervention and control groups where only the active intervention groups will receive weekly contact for the first 3 months to help meet intervention targets. As a result, intervention completers could be more motivated than control participants which may confound results. Lastly, consistent with other lifestyle interventions, participants will not be blinded to treatment allocation; however, data will be analysed by a blinded member of the study team to minimise detection bias.

## CONCLUSION

PROMED-EX focuses on undernutrition and cognitive impairment which are major contributors to mortality and disability in older age. The findings will provide new and critical insight into how modifications of diet and exercise behaviour can act to prevent undernutrition in older citizens at risk of undernutrition and cognitive decline. The study will advance understanding of how the protein-enriched MedDiet influences nutritional status and cognitive performance, and whether biological pathways are altered by the addition of exercise. We expect the study findings to inform future dietary guidelines for prevention of undernutrition to promote healthy neurocognitive ageing, and guide the design of future larger trials to test the efficacy of diet and exercise on cognition.

**Author affiliations**
[1]Centre for Public Health, Queen's University Belfast, Belfast, UK
[2]School of Agriculture and Food Science, Institute of Food and Health and Conway Institute, University College Dublin, Dublin, Ireland
[3]Division of Human Nutrition, Wageningen University, Wageningen, The Netherlands
[4]Neuroscience Institute, Aging Branch, National Research Council, Padua, Italy
[5]ExWell Medical, Irish Wheelchair Association, Dublin, Ireland
[6]Institute for Biomedical Technologies, Epidemiology Unit, National Research Council, Segrate, Italy
[7]Department of Medical Sciences, University of Ferrara, Ferrara, Italy
[8]Institute for Biomedicine of Aging, Friedrich-Alexander-Universitat Erlangen-Nurnberg, Nuremberg, Germany
[9]Global Brain Health Institute, Trinity College Dublin, Ireland & University of California, San Francisco, California, USA

**Acknowledgements** The authors would like to thank the study participants, primary care and other community colleagues who are assisting with recruitment. They are also grateful to the PPI and TSC members for their support in the study. They also acknowledge the European Joint Programming Initiative 'A Healthy Diet for a Healthy Life' (JPI HDHL) and of the ERA NET Cofund ERA HDHL (GA N 696295 of the EU Horizon 2020 Research and Innovation Programme) for funding the overall 'PROMED-COG consortium project' including the national funding bodies: the UK Research & Innovation (UKRI) Biotechnology and Biological Sciences Research Council and Medical Research Council, the Italian Ministry of Universities and Research and the Health Research Board Ireland.

**Contributors** CTM, LB, CRC, FP, LCdG, BM, MCM, JW and DV contributed to the conception and design of the trial. CTM and NAW drafted the manuscript. All authors (CTM, NAW, RR-M, LB, CRC, LCdG, SM, NM, BM, MCM, MN, RO'N, FP, GS, CT, DV, JW) contributed to the editing of the manuscript and revising it critically for intellectual content. The Trial Steering Committee approved this manuscript submission. CTM gave final manuscript approval and is the corresponding and senior author.

**Funding** PROMED-EX trial was funded by Biotechnology and Biological Sciences Research Council (Grant Ref: BB/V019201/1).

**Competing interests** None declared.

**Patient and public involvement** Patients and/or the public were involved in the design, or conduct, or reporting, or dissemination plans of this research. Refer to the Methods section for further details.

**Patient consent for publication** Not required.

**Provenance and peer review** Not commissioned; externally peer reviewed.

terminology, drug names and drug dosages), and is not responsible for any error and/or omissions arising from translation and adaptation or otherwise.

**ORCID iD**
Nicola Ann Ward http://orcid.org/0000-0002-4591-8443

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
