## [Reviewer comments · BMJ Open]

ARTICLE DETAILS

TITLE (PROVISIONAL)	Study Protocol: Randomised controlled trial to evaluate the effects of a PROtein enriched MEDiterranean Diet and EXercise on nutritional status and cognition in undernourished adults with subjective cognitive decline: The PROMED-EX Study.
AUTHORS	Ward, Nicola Ann; Reid-McCann, Rachel; Brennan, Lorraine; Cardwell, Christopher; de Groot, LC; Maggi, Stefania; McCaffrey, Noel; McGuinness, Bernadette; McKinley, Michelle C.; Noale, M; O'Neill, Roisin; Prinelli, Federica; Sergi, Giuseppe; Trevisan, Caterina.; Volkert, Dorothee; Woodside, Jayne; McEvoy, Claire T.

VERSION 1 – REVIEW

REVIEWER	Geirdottir, Olof University of Iceland School of Health Sciences, Faculty of Food Science and Nutrition
REVIEW RETURNED	14-Feb-2023

GENERAL COMMENTS	The intruding study, clear study protocol and ambitious aims make the readers hope for exciting results, which makes this paper interesting. Introduction Line 13-15; authors make claims about the causal effect that weight loss increases the risk of dementia. However, prospective studies show association, not causal effects, as the authors say in line 20. The introduction is well-written, and the literature is up to date. Methods and analysis Could the Sample size part (page 13, line 26) be part of the study design (page 4, line 44), so the reader will not wonder why n=105 for several pages Intervention Line 36; Daily energy requirements will be calculated at 30 kcal/kg with a supplement of 400-600 kcal only if weight gain is indicated. Could authors clarify what will be done if weight gain is not shown? Discussion Well written
--

REVIEWER	Macpherson, Helen Deakin University Faculty of Health
REVIEW RETURNED	28-Feb-2023

GENERAL COMMENTS	Thank you for the opportunity to review this paper which provides a protocol for a 3 arm RCT investigating the effects of a protein enriched diet (+ or –) a home based exercise program compared to standard care in older, underweight people with SCD. The study
---

	consists of a well-considered range of outcome measures, intervention duration and delivery. The personalised focus on the intervention is likely to have real world benefit. However, there seems to be an over emphasis on the perceived novelty of the intervention and a lack of consideration regarding factors from other comparable studies which have led to their success or null findings. The emphasis on the secondary outcome of cognition also seems to be over-stated given that no sample size calculation is provided for these outcomes and the sample size is likely to be underpowered to detect any cognitive effects in this sample. Below are some further considerations: Background Paragraph 3, - “However, small sample sizes as well as differences in study populations and nutritional interventions (dietetic counselling,[23] oral supplementation,[24,25] meal provision[25]), and cognitive outcomes make it difficult to know whether addressing undernutrition could protect against cognitive decline” What about residential status – aged care / community dwelling as a an important influence regarding heterogeneity in study findings? The background should cover previous studies which have combined protein with exercise for cognition (i.e RCTS have examined meat or supplements) and justify why the proposed approach may lead to better outcomes. Whilst a protein enriched diet may be beneficial for nutritional status, the literature regarding increasing protein intake for cognition is very mixed. What is the rationale for focusing on SCD? This should be covered in the introduction Method: Generally individuals who participate in diet / exercise trials for cognitive function / dementia prevention are healthier than the general population. How will the researchers aim to capture individuals who are underweight and also have memory concerns? Diet intervention: It is unclear whether a “trained researcher” is appropriate to provide personalised dietary advice. Ideally this should be led by a dietitian, what training will the researcher have received? The same query applies to the researcher conducting the weekly phone calls. What is the evidence that participants will reach the increased protein target goal whilst self-selecting from the foods delivered? Similarly who is prescribing the exercise program and what are their required qualifications? It is not clear whether all participants will receive weekly phone calls during the first 3 months or only the interventions groups? If there is a difference in the amount of social contact for each group with the researchers this would be considered a confounding influence. A large number of cognitive tests are listed, will there be a composite measure calculated for global cognition? Will there be statistical correction for multiple cognitive measures used? The study aims cover both nutritional status and cognition, yet cognition is listed as a secondary outcome. The sample size calculation is based on the MNA alone. The authors should consider whether to reduce the focus on cognition of this study, given it is likely very underpowered to detect a cognitive effect of the intervention (potential 4 levels of an unsupervised exercise
--	---

	program is likely to lead to heterogenous outcomes). Although the dietary intervention uses strategies that have previously been successful to improve adherence to the MedDiet, if participants are underweight at baseline, there may need to be a large number of goals for participants to work towards. What are the planned covariates to be included in the ITT analysis? The primary outcome is the between groups difference in MNA score but the sample size is based on an increase in MNA of 3 points for the 2 intervention groups. Therefore the primary outcome analysis is only powered to identify a difference between the interventions and standard care, not between the two interventions. The inclusion of two intervention arms are then likely only to have differential effects on secondary outcomes. This could be more clearly stated and also reiterated in the limitations in the discussion (limited power to detect between intervention arm effects). Discussion: The discussion does not address how this project will advance the area / build on recent RCTs. The statements in the first 2 paragraphs are largely unsupported by any literature. The only novel components of this study are adjustment of the MedDiet to be higher in protein and the recruitment of underweight participants. These adjustments to the study design should be better interpreted with regards to previous RCTs. “Prior studies have relied on the limited MMSE global cognition to measure cognitive performance” This statement is incorrect and ignores a large literature of exercise and diet interventions for cognition, even in individuals with SCD. The impact of partial blinding on the study results could be further elaborated, i.e. reduce effect size as individuals in the standard care group also make some positive diet changes.
--	--

VERSION 1 – AUTHOR RESPONSE

Reviewer: 1 Prof. Olof Geirsdottir, University of Iceland School of Health Sciences		
The intruding study, clear study protocol and ambitious aims make the readers hope for exciting results, which makes this paper interesting.	The authors thank reviewer #1 for their positive feedback.	N/A
Introduction Line 13-15; authors make claims about the causal effect that weight loss increases the risk of dementia. However, prospective studies show association, not causal effects, as the authors say in line 20. The introduction is well-written, and the literature is up to date.	We thank the review for raising this important point and agree with the comment. The prospective studies do suggest an association between weight loss and dementia risk and as discussed in line 20 there is the issue of reverse causality.	Introduction; Paragraph 1 ; Line 14-15.

	The sentence has been amended to reflect this. “In prospective studies, weight loss has been associated with increased dementia risk,[6] and precedes the onset of dementia by a decade or more,[7,8] making it an intervention target that should be prioritised.”	
Methods and analysis Could the Sample size part (page 13, line 26) be part of the study design (page 4, line 44), so the reader will not wonder why n=105 for several pages	We thank reviewer 1 for this helpful suggestion. To address this point, we have moved the sample size section. These changes can be seen in Methods and Analysis; paragraph 4.	Methods and analysis; Paragraph 4 (sample size.)
Intervention Line 36; Daily energy requirements will be calculated at 30 kcal/kg with a supplement of 400-600 kcal only if weight gain is indicated. Could authors clarify what will be done if weight gain is not shown?	We thank reviewer 1 for this query and wish to highlight that our primary outcome is between group differences in overall nutritional status as measured on the MNA. Weight gain is not an outcome measure per se and many of our participants at nutritional risk will not need to aim for weight gain. The MNA score considers recent changes in food intake, appetite, recent weight loss and BMI. Individuals will be counselled on weight gain only if indicated at baseline and will be monitored via weekly telephone calls with advice being altered accordingly to increase energy and protein intake. If weight gain occurs in response to intervention this is will be reflected in the total MNA score.	N/A
Discussion Well written	The authors thank the reviewer for the positive feedback regarding the discussion.	N/A
Reviewer: 2 Dr. Helen Macpherson, Deakin University Faculty of Health		
Thank you for the opportunity to review this paper which provides a protocol for a 3 arm RCT investigating the effects of a protein enriched diet (+ or –) a home based exercise program compared to standard care in older, underweight people with	The authors thank reviewer 2 for taking the time to review our protocol paper and for the positive feedback. We have tried to address all points below but please note some reviewer suggestions could not be incorporated.	N/A

SCD. The study consists of a well-considered range of outcome measures, intervention duration and delivery. The personalised focus on the intervention is likely to have real world benefit. However, there seems to be an over emphasis on the perceived novelty of the intervention and a lack of consideration regarding factors from other comparable studies which have led to their success or null findings. The emphasis on the secondary outcome of cognition also seems to be over-stated given that no sample size calculation is provided for these outcomes and the sample size is likely to be underpowered to detect any cognitive effects in this sample. Below are some further considerations	Novel aspects of the study include the Mediterranean diet approach enriched with protein to meet higher needs of target older adults. No prior studies have evaluated the effects of this optimised dietary approach combined with exercise on cognitive performance. Furthermore, prior multidomain studies have recruited adults at increased risk of cardiovascular disease or dementia using a risk prediction score. Uniquely, we have focused on older adults at risk of undernutrition who report SCD. This vulnerable population has not yet been studied extensively in the lifestyle-dementia field. A further novelty aspect lies particularly in the metabolomic and biomarker analysis that will allow the exploration of mechanistic pathways for diet and exercise induced change in nutritional status and cognition that are yet unknown in this high-risk population. The study is powered to see an effect on MNA as the primary outcome and then gather secondary outcomes on cognitive performance on which to base further studies.	
Background Paragraph 3, - "However, small sample sizes as well as differences in study populations and nutritional interventions (dietetic counselling,[23] oral supplementation,[24,25] meal provision[25]), and cognitive outcomes make it difficult to know whether addressing undernutrition could protect against cognitive decline" What about residential status – aged care / community dwelling as a an important influence regarding heterogeneity in study findings?	The authors agree with review #2 and have clarified this in the text. 'However, small sample sizes as well as difference in residential status of study populations and nutritional interventions....'	Introduction; Paragraph 3
The background should cover previous studies which have combined protein with exercise for	We appreciate the reviewer's suggestion but we prefer to focus on prior studies targeting undernutrition	N/A

cognition (i.e RCTS have examined meat or supplements) and justify why the proposed approach may lead to better outcomes. Whilst a protein enriched diet may be beneficial for nutritional status, the literature regarding increasing protein intake for cognition is very mixed.	for cognitive benefit as our primary outcome is focused on nutritional status and this is the novel aspect of the trial.	
What is the rationale for focusing on SCD? This should be covered in the introduction	We thank the reviewer for the question and believe we have covered this sufficiently in our introduction. Studies have previously looked at the efficacy of dietary interventions for cognition in either healthy or undernourished dementia patients. As SCD is a self-reported experience of memory loss and potentially one of the earliest noticeable symptoms of AD/related dementia, the proposed dietary and exercise intervention may be most efficacious in this unique study population. Please see the rationale taken from the introduction. “Trials are warranted to test the efficacy of intervention for undernutrition in populations at increased risk of cognitive decline but not yet cognitively impaired and may benefit from early intervention.”	Introduction; paragraph 3.
Method: Generally individuals who participate in diet / exercise trials for cognitive function / dementia prevention are healthier than the general population. How will the researchers aim to capture individuals who are underweight and also have memory concerns?	We thank the reviewer for the query. We have a strong track record of successfully recruiting individuals to take part in lifestyle studies. Several strategies have been implemented to capture individuals from this vulnerable population including GP referral, identification of potentially eligible participants from community networks, primary care sectors and health and social care trusts and more. We have outlined our recruitment methods in the protocol [see page 5 ‘Recruitment and Eligibility of Study Participants’]	N/A

	The eligibility screening process will exclude those not at risk of undernutrition and SCD.	
Diet intervention: It is unclear whether a “trained researcher” is appropriate to provide personalised dietary advice. Ideally this should be led by a dietitian, what training will the researcher have received? The same query applies to the researcher conducting the weekly phone calls	We thank the reviewer for the query regarding the dietary intervention. The PROMED-EX Trial is overseen by a registered dietitian. The trained researchers have background in nutrition and exercise and have undergone competency training to deliver the PROMED diet and Exwell Medical exercise programme [with the Exwell programme collaborators on the study]. We have a standardised operating procedure for delivery of the intervention and follow-up calls for both PROMED-EX (diet & exercise) and PROMED (diet only), written checklists with questions are used and relevant information is collected. In addition, the team has weekly meetings to discuss any questions or queries that might come up.	N/A
What is the evidence that participants will reach the increased protein target goal whilst self-selecting from the foods delivered?	The strategies that we are using have been successful in achieving diet behaviour change in community-based populations including older adults and are outlined in the protocol ‘Maximising and monitoring compliance.....’ [page 8] To clarify, baseline protein intake is calculated from a 4-day food diary and individuals receive tailored advice to meet the intervention protein target. Participants are encouraged to make small, easy to measure and achievable goals, where the trained researcher discusses how these goals will help to reach the protein target e.g. incorporation of lean protein (chicken, turkey, fish), legumes, wholegrains varieties, natural nuts (and more). The preferred food items to meet protein	N/A

	[and dietary] goals will be delivered to the home as outlined in the protocol [page 6-7 'Protein enriched MedDiet intervention'] Weekly calls with the trained researcher will support participants towards their protein targets.	
Similarly who is prescribing the exercise program and what are their required qualifications?	Thank you- please see the earlier response regarding "trained researchers."	N/A
It is not clear whether all participants will receive weekly phone calls during the first 3 months or only the interventions groups? If there is a difference in the amount of social contact for each group with the researchers this would be considered a confounding influence.	We confirm that the levels of contact will differ between the intervention and control groups. where only the active intervention groups will receive weekly contact for the first 3 months to help meet the intervention goals to improve MNA score. At this point intervention completers are likely to be more motivated than control participants which can confound results. For the remaining 3 months there is no scheduled support for intervention participants. We will discuss any limitations of the study design in the context of results in a future manuscript.	N/A
A large number of cognitive tests are listed, will there be a composite measure calculated for global cognition? Will there be statistical correction for multiple cognitive measures used?	Secondary endpoints are exploratory and the raw data will be cleaned and summarised. We intend to create summary composite scores for global cognition and for domains of memory and executive function. This has been clarified in the text [page 10 ' cognitive function ']. Given the exploratory nature of the secondary data we don't anticipate statistical correction, but our medical statistician will advise whether this will be required.	Page 10 paragraph 4
The study aims cover both nutritional status and cognition, yet cognition is listed as a secondary outcome. The sample size calculation is based on the MNA alone. The authors should consider whether to reduce the focus on cognition of this study, given it is	The study is powered for effect on MNA as the primary outcome. Due to the novelty of this intervention and study population, it was not possible to do a sample size calculation for cognition however the findings of PROMED-EX on cognitive performance will be used to calculate	N/A

likely very underpowered to detect a cognitive effect of the intervention	sample size and inform the design future studies.	
(potential 4 levels of an unsupervised exercise program is likely to lead to heterogenous outcomes). Although the dietary intervention uses strategies that have previously been successful to improve adherence to the MedDiet, if participants are underweight at baseline, there may need to be a large number of goals for participants to work towards.	Participants are at increased nutritional risk rather than underweight and, like the dietary component, the prescription of the exercise programme will be tailored to baseline fitness and employ behavioural strategies to allow goal setting and self-monitoring of performance During the weekly calls, participants are asked (based on the Borg scale) their rate of perceived exertion and whether they were able to achieve their exercise goals. This is used to tailor the programme for easier or harder levels. All participants aim for somewhat hard, and this is recorded in their personal exercise logs. Compliance with the intervention will be examined at the end.	N/A
What are the planned covariates to be included in the ITT analysis?	Randomisation should largely mean the groups are similar with respect to known and unknown characteristics therefore there are no a priori plans to include covariates.	N/A
The primary outcome is the between groups difference in MNA score but the sample size is based on an increase in MNA of 3 points for the 2 intervention groups. Therefore the primary outcome analysis is only powered to identify a difference between the interventions and standard care, not between the two interventions. The inclusion of two intervention arms are then likely only to have differential effects on secondary outcomes. This could be more clearly stated and also reiterated in the limitations in the discussion (limited power to detect between intervention arm effects)	We thank reviewer for this comment and agree. We have not powered to detect a difference between the two intervention groups as it is difficult to anticipate what kind of difference we might get and would want to detect as statistically or clinically significant. We could also detect a difference of 3 between the 2 intervention groups e.g. if both groups increased MNA by 3 we would not detect this as statistically significant but if one group improved by 1 and the other by 4 we would. We have added this to the limitations section of the discussion [page 13, paragraph 4].	Page 13 Paragraph 4
Discussion: The discussion does not address how this project will	We thank reviewer #2 for highlighting some of the novel aspects of the trial.	N/A

advance the area / build on recent RCTs. The statements in the first 2 paragraphs are largely unsupported by any literature. The only novel components of this study are adjustment of the MedDiet to be higher in protein and the recruitment of underweight participants.	Aside from the protein-enriched Mediterranean diet intervention, and unique study population of older adults at risk of undernutrition with subjective memory decline. The integration of state-of-the-art metabolomics, metabolic, inflammatory, and nutritional biomarkers will help elucidate biological mechanisms underlying how diet can influence nutrition status and cognition and how exercise can interact with nutrition to modify these processes. Furthermore, the planned evaluation will provide insight into the feasibility of lifestyle behaviour changes in a high-risk population.	
These adjustments to the study design should be better interpreted with regards to previous RCTs. "Prior studies have relied on the limited MMSE global cognition to measure cognitive performance" This statement is incorrect and ignores a large literature of exercise and diet interventions for cognition, even in individuals with SCD.	We thank the reviewer for pointing out this error and the sentence has been rephrased. "Prior studies in undernourished patients have relied on the limited MMSE global cognition to measure cognitive performance.[25] While MMSE is a validated screener for cognitive impairment, it provides limited examination of visuospatial cognitive ability, long-term memory, and language functions. It also has a "ceiling" effect whereby individuals can still score quite highly even though they may have significant cognitive impairment.[70]" Please see an additional sentence added acknowledging the use of cognitive test battery in FINGER trial.	Discussion; paragraph 2
The impact of partial blinding on the study results could be further elaborated, i.e. reduce effect size as individuals in the standard care group also make some positive diet changes.	We have acknowledged that participants will not be blinded to treatment allocation [page 14] and if there are unanticipated effects in the control group we will discuss this in a future manuscript.	Discussion; paragraph 4